# Influence of Mo and Fe on Photosynthetic and Nitrogenase Activities of Nitrogen-Fixing Cyanobacteria under Nitrogen Starvation

**DOI:** 10.3390/cells11050904

**Published:** 2022-03-05

**Authors:** Asemgul K. Sadvakasova, Bekzhan D. Kossalbayev, Aziza I. Token, Meruert O. Bauenova, Jingjing Wang, Bolatkhan K. Zayadan, Huma Balouch, Saleh Alwasel, Yoong Kit Leong, Jo-Shu Chang, Suleyman I. Allakhverdiev

**Affiliations:** 1Faculty of Biology and Biotechnology, Al-Farabi Kazakh National University, Al-Farabi Avenue 71, Almaty 050038, Kazakhstan; asem182010@gmail.com (A.K.S.); t.aziza_93@mail.ru (A.I.T.); bauyen.meruyert@gmail.com (M.O.B.); zbolatkhan@gmail.com (B.K.Z.); huma@comsats.org (H.B.); 2Department of Chemical and Biochemical Engineering, Geology and Oil-Gas Business Institute Named after K. Turyssov, Satbayev University, Almaty 050043, Kazakhstan; 3Tianjin Institute of Industrial Biotechnology, Chinese Academy of Sciences, No. 32, West 7th Road, Tianjin Airport Economic Area, Tianjin 300308, China; wang_jj@tib.cas.cn; 4College of Science, King Saud University, Riyadh 12372, Saudi Arabia; salwasel@ksu.edu.sa; 5Department of Chemical and Materials Engineering, Tunghai University, Taichung 407, Taiwan; yoongkitleong1014@gmail.com (Y.K.L.); changjs@mail.ncku.edu.tw (J.-S.C.); 6Research Center for Smart Sustainable Circular Economy, Tunghai University, Taichung 407, Taiwan; 7Department of Chemical Engineering, National Cheng Kung University, Tainan 701, Taiwan; 8Controlled Photobiosynthesis Laboratory, K.A. Timiryazev Institute of Plant Physiology, Russian Academy of Sciences, Botanicheskaya Street 35, 127276 Moscow, Russia

**Keywords:** cyanobacteria, heavy metals, photosynthesis, nitrogenase, heterocyst

## Abstract

The potential of cyanobacteria to perform a variety of distinct roles vital for the biosphere, including nutrient cycling and environmental detoxification, drives interest in studying their biodiversity. Increasing soil erosion and the overuse of chemical fertilizers are global problems in developed countries. The option might be to switch to organic farming, which entails largely the use of biofertilisers. Cyanobacteria are prokaryotic, photosynthetic organisms with considerable potential, within agrobiotechnology, to produce biofertilisers. They contribute significantly to plant drought resistance and nitrogen enrichment in the soil. This study sought, isolated, and investigated nitrogen-fixing cyanobacterial strains in rice fields, and evaluated the effect of Mo and Fe on photosynthetic and nitrogenase activities under nitrogen starvation. Cyanobacterial isolates, isolated from rice paddies in Kazakhstan, were identified as *Trichormus variabilis* K-31 (MZ079356), *Cylindrospermum badium* J-8 (MZ079357), *Nostoc* sp. J-14 (MZ079360), *Oscillatoria brevis* SH-12 (MZ090011), and *Tolypothrix tenuis* J-1 (MZ079361). The study of the influence of various concentrations of Mo and Fe on photosynthetic and nitrogenase activities under conditions of nitrogen starvation revealed the optimal concentrations of metals that have a stimulating effect on the studied parameters.

## 1. Introduction

The most important direction in the development of organic agriculture is the creation of microbial biotechnologies that help maintain fertility and intensify agricultural production. Soil fertility is directly related to the diversity, quantity, and activity of the soil microbiota, which determine the transformation, migration, and accumulation of substances in the soil ecosystem, and provide plants with all necessary assailable nutrients. At the same time, it is known that the increase in crop yields largely depends on access to mineral nutrients, especially nitrogen. The natural sources of nitrogen in the soil are microorganisms, which can fix molecular nitrogen from the atmosphere [1,2,3,4].

In this context, one of the most interesting groups of microorganisms are cyanobacteria, which, in addition to the ability to fix nitrogen, have a wide range of adaptations to different soil and hydrothermal conditions [5,6,7]. Herewith, both types of cyanobacteria (with heterocysts and without them) have the ability to fix nitrogen. However, in heterocystous cyanobacteria, the processes of photosynthesis and nitrogen fixation occur simultaneously and independently of the time of day, while in non-heterocystous cyanobacteria, these processes occur separately: nitrogen fixation, at night; photosynthesis during the day [8]. In the cyanobacterial species that form heterocysts, the nitrogenase is located in the heterocyst and is, thus, protected from the inhibitory effects of oxygen [9,10,11,12,13]. They are more attractive in terms of productivity and can enrich the soil with 20–30 kg N ha^−1^ per year, which is a great economic advantage for agriculture. In addition, they are technologically advanced, can grow on cheap media without sources of organic compounds and mineral nitrogen, and do not require expensive equipment [14,15]. Considering the main factors affecting nitrogenase activity, it is worth outlining the role of biogenic metals in nitrogenase enzyme cofactors that determine the intensity of nitrogen fixation by free-living cyanobacteria [16,17].

Despite the large amount of data available in the literature indicating the high metabolic potential of cyanobacteria in nitrogen fixation, there are still some problems related to their productivity that limit their widespread use in agro-biotechnology. Therefore, the attention given to this group of organisms is focused on the study of the basic cultivation conditions that increase their nitrogen fixation abilities.

In this regard, scientific research aimed at potentially enhancing the nitrogen-fixing ability of cyanobacteria is currently very relevant. Primarily, research in this area should focus on finding new, more productive strains of nitrogen-fixing cyanobacteria and developing new approaches to increase the efficiency of their metabolic activity.

Therefore, this work aims to select and study new strains of nitrogen-fixing cyanobacteria. The article demonstrates the results of isolation, investigation, and identification of five cyanobacterial strains isolated from rice paddies in Kazakhstan and the influence of Mo and Fe on their photosynthetic and nitrogenase activity.

## 2. Materials and Methods

### 2.1. Research Objects

Five different strains of cyanobacteria, including *Nostoc* sp. (J-14), *Cylindrospermum* sp. (J-8), *Anabaena variabilis* (K-31), *Oscillatoria brevis* (SH-12), and *Tolypothrix tenuis* (J-1), were isolated from the Kyzylorda region. The strains were isolated from the rice fields and aquatic ecosystems of the Zhanakorgan and Shieli districts of the Kyzylorda region (rice paddies of Avangard, Akzhol, and Zhayilma villages, and water basins adjacent to rice fields, i.e., the tributaries of the Syr-Darya river: Kuday Kul, Zhayilma).

### 2.2. Isolation and Cultivation of Cyanobacterial Strains

Eighteen water samples and twenty-six composite samples of soil and cyanobacterial mats collected from the surface of plants using classical algological methods served as material for this study. Identifiers of freshwater microalgae and cyanobacteria were used to determine the species. In order to obtain enrichment cultures of cyanobacteria, 5 mL of aliquot samples were placed in 100 mL flasks containing 5 mL of BG-11 [18], Allen [19], Bold basal liquid [20] culture medium with the addition of cycloheximide (Biochem Chemopharma, Cosne-Cours-sur-Loire, France) at a concentration of 50 μg mL^−1^ to inhibit the growth of eukaryotes. Simultaneously, samples were inoculated into Petri dishes with solid medium (1% agar) BG-11. Cultivation was carried out in parallel at temperatures of 18–22 °C with a constant illumination of 50 μmol (photons) m^−2^ s^−1^. Unialgal cultures of cyanobacteria were obtained through the periodic inoculation of the enrichment culture on solid nutrient medium BG-11, Allen, Gromov, and Bold, using capillary pipettes to separate the cyanobacterial filaments from the mixed microbial suspension. Several antibiotics (penicillin, gentamicin, tetracycline, neomycin, ampicillin, chloramphenicol, and kanamycin) at different concentrations from 1500 unit (U) mL^−1^ to 25,000 U mL^−1^ and their combinations with a total concentration of 20,000 U mL^−1^ were used to purify isolated cyanobacterial cultures from satellite bacteria. The calculated concentrations of antibiotics were added to the culture media after sterilization. The strains of isolated cyanobacteria were cultured on liquid and agar nutrient media of standard composition BG-11 and Allen in flasks in the luminostat. Preliminary morphological identification was performed under a light microscope (Micro Optix C600, Wiener Neudorf, Austria). Morphological identification was based on some characteristic morphological features, such as cell size, shape, location of cells, presence or absence of mucus, etc., using taxonomic literature and various photo galleries. The determination of the species composition of phototrophic eukaryotes was based on morphological and cultural characteristics using various determinants [21,22]. Before conducting an experiment, the cyanobacterial culture was subcultured from the liquid medium with the stored culture on Petri dishes with a freshly prepared nutrient medium of the same composition.

### 2.3. Molecular Identification of Cyanobacterial Strains

Cyanobacterial species were identified using molecular genetic research methods by sequencing the conserved DNA locus. To accomplish the tasks assigned, the following steps were performed: the isolation of genomic DNA, sample preparation and polymerase chain reaction, DNA sequencing for a fragment of the conserved locus, and an analysis of the nucleotide sequence of the conserved 16S rRNA locus [23].

#### 2.3.1. Isolation of Genomic DNA of Cyanobacteria

Genomic DNA was isolated following the protocol proposed by Dale et al. [24]. Cyanobacterial cells were pelleted via centrifugation at 4000× *g* for 5 min, the supernatant was removed, and the cells were suspended in 1 mL of extraction buffer with the following composition: 200 mmol Tris-HCl (pH-8.0), 400 mmol LiCl, 25 mM EDTA, and 1% SDS. Acid-treated glass beads with a diameter of 425–600 μm (Sigma-Aldrich, Darmstadt, Germany) were added to the suspension. The resulting suspension was shaken vigorously for 30 s and then incubated on ice for 30 s. This procedure was repeated 5 times. The resulting mixture was centrifuged at 3000× *g* for 15 min at 4 °C. The supernatant was then transferred to a sterile microtube containing 600 μL of ice-cold isopropanol and mixed by pipetting, then centrifuged at 16,000× *g* for 20 min at 4 °C. The precipitate was dissolved in 100 μL TE buffer (2 mmol Tris-HCl (pH-8.0), 1 mmol EDTA). Then, 10 μL 3 M sodium acetate (pH-5.2) and 300 μL of 96% ethanol were added and incubated at −20 °C for 2 h. The precipitate was centrifuged. Another round of centrifugation was performed for 30 min with 16,000× *g* at 4 °C. The pellet was washed with 70% ethanol, oven-dried, and dissolved in 50 μL TE buffer. DNA concentration was determined using a Nanovue plus spectrophotometer (Biochrom, Boston, MA, USA). The concentration of isolated genomic DNA was adjusted to 100 ng μL^−1^.

#### 2.3.2. Sample Preparation and Polymerase Chain Reaction

The ribosomal DNA gene was selected as a conservative DNA locus for the cyanobacterial identification. The amplification of a 16S rRNA gene fragment from bacterial genomic DNA was performed using primers: 8f, 806r, M23f, M23′f, 27f, 1492r, 1525r [25,26,27]. The separation of the amplification products was carried out via DNA electrophoresis using 1% agarose gel in TAE buffer containing ethidium bromide (15 μg mL^−1^). The resulting PCR products were treated with a mixture of Exol exonuclease and FastAP alkaline phosphatase to remove the remaining primers and dNTPs. To 10 μL of each amplicon, 1 μL of Exol and FastAP was added and incubated for 30 min at 37 °C. The inactivation of enzymes was carried out at 85 °C for 15 min [28].

#### 2.3.3. DNA Sequencing Using the Fragment of the Conserved Locus

DNA sequencing using the Sanger method was used to determine the nucleotide sequence of the conserved locus [29]. The analysis of chromatograms and their comparison with the reference sequence was carried out using the VectorNTI version 11 software package and the NCBI database using the BLAST (Basic Local Alignment Search Tool) service [30].

### 2.4. Methods for Quantification of Microalgal Cells and Determination of Biomass Growth Productivity

The change in the concentration of cyanobacterial biomass was determined by measuring the optical density on a PD-303UV spectrophotometer (Apel, Saitama, Japan) at a wavelength of 540–590 nm. The biomass productivity was determined according to Tsarenko et al. [31]. The cells were precipitated using a 5810R centrifuge (Eppendorf, Hamburg, Germany) at 5000× *g* for 5 min at 25 °C. The growth rate coefficient of the cyanobacterial cultures was calculated based on the increase in the cell number in the experimental vessels, taking into account the initial cell number and the cell number after a certain time.

### 2.5. Determination of the Heterocyst Formation Frequency

The frequency of heterocyst formation was analysed on nitrogen-free BG_0_-11 [32] medium, and the control was a BG-11 nutrient medium with a standard composition. Before the experiment, the cultures were kept in a stationary phase without nutrients for 3 days to minimize the effects of the previous BG-11 starting medium on the growth of the cyanobacteria studied. Then, the cyanobacterial cells were centrifuged in the 5810R centrifuge at 1000× *g* for 10 min and the pellets were washed twice in saline solution to remove residual N and P from the cell surface. Cells were then incubated in flasks containing two versions of fresh media—BG-11 and BG_0_-11 without N—at the same temperature and light intensity as described above. The frequency of heterocyst appearance was recorded daily for 9 days under a light microscope (Micro Optix OPTIX C600, Wiener Neudorf, Austria) at magnifications of 200 and 400. For this purpose, 4 mL of the sample was taken with a micropipette, and a crushed droplet was prepared and analysed. Heterocysts were identified by their thickened cell wall, pale colour, and the formation of the poles compared to neighbouring cells [33]. The frequency of heterocysts was calculated as the percentage of heterocyst density to the total density of vegetative cells when at least 10 fields of view were counted per sample. Thus, the number of vegetative cells per heterocyst was determined for 10 heterocysts, and the data are expressed as mean ± standard deviation. If there were not enough cyanobacterial cells to count 10 heterocysts, the data were noted as too small to count.

### 2.6. Determination of Photosynthetic Activity

Chlorophyll fluorescence parameters were measured using a Multi-function Plant Efficiency Analyzer (M-PEA-2, Hansatech Instruments, King’s Lynn, UK) after 24 h of cultivation. The following fluorescence parameters were recorded during the experiments: F_0_ is the fluorescence value at open reaction centres, and F_m_ is the maximum fluorescence after a series of light flashes saturate the photosynthesis reaction centres. Before measurement, samples were kept in the dark for 15 min. These measured quantities were used to calculate the following parameters [34,35]:

F_V_ = F_m_ − F_0_ is the maximum variable fluorescence;

F_V_/F_m_ is the maximum quantum yield of PSII of the primary photochemical reaction in open PSII reaction centres: F_V_/F_m_ = φPo.

All measurements were carried out in at least five repetitions. The figures show data of average values.

### 2.7. Determination of Nitrogenase Activity Using the Acetylene Method

The cultures of cyanobacteria were cultivated in the light until the beginning of the stationary phase of the cells. Then, they were centrifuged at 16,000× *g* for 20 min at 4 °C. The obtained biomass was adjusted to OD_720_ = 1.5, then 15 mL thick biomass was transferred to 25 mL GC vial. Next, a gas mixture of 10% acetylene and 90% argon was introduced into the GC vial within 30 min [36]. The cells inside the vial were cultured for 1, 2, 4, 8, 24, and 32 h at a photosynthetic photon flux density of 70 μmol (photons) m^−2^ s^−1^. After incubation, 500 μL gas samples were withdrawn and the concentration of ethylene in the gas mixture was determined. The redox activity of acetylene was determined on gas chromatograph GC-15A (Shimadzu, Kyoto, Japan) in the form of nmol ethylene mg^−1^ DW (dry weight) h^−1^.

### 2.8. Effect of Different Concentrations of Mo and Fe on Nitrogenase Activity

The influence of these metals on the cultures was studied in Allen’s medium. Of the metals studied, Mo (0.017 μmol) is present in this medium as a trace element in trace amounts and Fe (0.36 μmol) is present in an insignificant concentration. The study of the stimulatory effect of these metals on nitrogenase and photosynthetic activities was carried out with their additional introduction at different concentrations. Both metals were not excluded from the medium because they are of crucial importance, since Fe, in addition to the process of nitrogen fixation, is also necessary for cell growth, in particular for the processes of photosynthesis and respiration.

Salt (FeCl_3_ × 6H_2_O) was added to the nutrient medium in concentrations of 1, 5, 10, and 15 µmol and Na_2_MoO_4_ in concentrations of 0.1, 0.5, and 5 µmol. The control for all variants of the experiment was the Allen_0_ medium (without nitrogen source), which contained Mo and Fe in concentrations equal to the composition of the standard medium. The cyanobacterial cultures were first cultured in Allen’s medium for 7 days under light at a temperature of 26 °C, then the cultures were transferred to Allen_0_ medium, which also contained various concentrations of metals, and then the strains were cultured for an additional 7 days until anaerobic conditions were established and the amount of ethylene was recorded. Nitrogenase activity was determined after 1, 2, 4, 8, 24, and 32 h.

### 2.9. Pigment Content

To determine the quantitative composition of pigments, cyanobacterial isolates were cultured for 14 days in Allen and Allen_0_ media, as well as in experimental variants with added metals, and then pigment content was analysed. From each variant analysed, 4 mL of a suspension of 14-day cyanobacterial culture was taken and then centrifuged at 15,000× *g* for 7 min, and the supernatant was discarded. Then, 4 mL of pre-chilled (+4 °C) methanol was added. The sample was homogenized by stirring (Silamat S6, 2 s), shaking (2000× *g*, 4 s), and pipetting. Then, the samples were covered with aluminium foil and incubated at +4 °C for 20 min to extract pigments from the cells. Next, the samples were centrifuged at 15,000× *g*, +4 °C for 7 min and the precipitate was discarded. The content of pigments (chlorophyll a, carotenoid, and phycocyanin) was determined using a spectrophotometer at wavelengths (A) of 470, 615, 650, 652, 665, and 720 nm using methanol as a control [37,38,39].

The concentration of the pigment content was calculated using Equations (1–3):Chlorophyll a (µg m^−1^) = 12.9447 (A_665_ − A_720_)(1)
Carotenoid (µg mL^−1^) = 1000 (A_470_ − A_720_) − 2.86(2)
Phycocyanin (µg mL^−1^) = (A_615_ − 0.474A_652_)/5.34(3)

### 2.10. Statistical Analysis

The studies were conducted in 3–5 repetitions. The statistical data processing was conducted using the RStudio software (version 1.3.959, R Studio PBC, 2020). An ANOVA was used for the evaluation of significance. When the ANOVA proved a significant difference, Tukey’s HSD test was performed to compare. Based on Tukey’s HSD test, treatments were categorized (into letters in descending gradation). To create graphs on the evaluation of the nitrogenase and photosynthetic activities of cyanobacteria isolated from rice paddies, Microsoft Office Excel 2013 was used. A phylogenetic tree was constructed using the neighbour-joining method in Mega software 6.0 with 1000 bootstrap values [40].

## 3. Results

### 3.1. Isolation of Cyanobacteria Axenic Cultures from Rice Fields

To study the biodiversity of phototrophic microorganisms and isolate nitrogen-fixing cyanobacteria, water, soil, algal mat, and plant samples were collected twice from rice fields at the end of the rice seedling and tillering phases. The study of rice fields in the Kyzylorda region, Republic of Kazakhstan (GeoCoordinate: 43°54′16″ N 67°15′04″ E), revealed a rich biodiversity of algoflora consisting of 58 species, forms and varieties of microalgae belonging to 5 phyla, 10 classes, 19 orders, 26 families, and 29 genera. The taxonomic structure of the studied algoflora is as follows: *Cyanobacteria*—25 (45%), *Bacillariophyta*—5 (10%), *Euglenophyta*—5 (3%), *Chlorophyta*—18 (33%), and *Charophyta*—5 (8%).

At the same time, it must be noted that there is a certain seasonality in the occurrence of the species studied. Thus, if at the end of the rice seedling phase there was a high frequency of occurrence of green algal species of the *Chlorophyta* and *Charophyta* phyla, which form a bright green mass on the surface and in the water layer, at the end of the tillering phase, they were replaced by representatives of the *Cyanobacteria* phylum, especially *Microcystis pulverea*, *Anabaena variabilis*, *Cylindrospermum* sp., *Merismopedia elegans*, *Gloeocapsa minuta*, *Stranonostoc linkia*, *Oscillatoria limosa*, *Spirulina major*, *Phormidium ambiguum*, and *Nostoc pruniforme*. They are found in the water layer and on the surface, on the leaves and stems of rice and weeds, and on the thallus of dying macroalgae. Representatives of diatoms were more common at the rice seedling phase.

To isolate cyanobacteria, samples of the algal mat were taken towards the end of the rice tillering phase. From the collected water, soil and algal mat samples and 18 enrichment cultures of the dominant species of phototrophic microorganisms were obtained on enriched media. Thirteen cyanobacterial and microalgal isolates, characterised by stable growth under laboratory conditions, were obtained using the serial passage method, nine of which were unialgal cultures.

Since the purpose of this work was to search for and study nitrogen-fixing cyanobacterial strains, our attention was focused only on the filamentous forms of the isolated strains. Thus, five promising isolates, namely SH-12, J-14, J-8, K-31, and J-1, had a filamentous structure and probably belonged to cyanobacteria.

The microscopic examination of these isolates showed that the isolates J-14, K-31, J-8, and J-1 belonged to the order of the *Nostocales* according to the morphological characteristics and differed from isolate SH-12 in the form and function of cell differentiation. Thus, the single, straight, sometimes slightly curved trichomes of these isolates consisted of blue-green, spherical or cylindrical, isodiametric vegetative cells, narrowed or not at the ends, with pronounced or slightly pronounced constrictions at stigma sites. A distinctive feature of these isolates was the formation of the terminal and intercalary heterocysts when the medium was exhausted on days 8–15 of cultivation; their size was not significantly different from that of the vegetative cells. In addition, the formation of akinetes was observed in the three isolates, J-14, K-31, and J-8, which were significantly different in size and shape from the vegetative cells, especially isolate J-8. They consisted of cylindrical or oval cells, were 5.1–7.7 × 2.6–3.4 μm in size, green, light-brown, yellow, granular and bordered on being heterocystous. The formation of akinetes was not observed in isolate J-1. In addition, this isolate was characterised by pronounced false branching of the trichomes, which made the culture appear bushy. The morphological characteristics of the fifth isolate SH-12 resembled those of the genus *Oscillatoria*, order *Oscillatoriales*, whose main feature was the non-differentiation of the trichomes. The culture consisted of homocytic, single, straight, single-rowed trichomes derived from small, homogeneous, cylindrical vegetative cells lying close together.

A detailed description of each isolate studied is given below (Figure 1):Culture J-14. Filamentous cyanobacteria. The trichomes are single, straight, not narrowed at the ends and have pronounced constrictions at the stigma sites and consist of blue-green spherical vegetative cells (length of 6–8 μm, width of 3–5 μm). The heterocysts are, in most cases, intercalary, solitary, and light-brown. Akinetes are rare, barely noticeably oval, their size does not differ from the size of the cells, and they are characterised by a granular content. They reproduce by hormogonia. Cultural characteristics: Grows well on Allen, Gromov, Bold, and BG-11 media at a temperature of 23–28 °C, with an initial pH of 7. At the same time, on a solid medium, prostrate growth is characteristic; on a liquid culture medium, growth is in the form of films both on the surface and at the bottom of the flask. In the old culture or on a medium with nitrogen deficiency, the development of heterocysts and akinetes is observed. According to the systematic position, it is classified as cyanobacteria, class *Hormogeneae*, order *Nostocales*, genus *Nostoc*, *Nostoc* sp.Culture J-8. Filamentous cyanobacteria. Cells form straight, loose, immobile (weakly motile) trichomes up to 15 μm in length, comprising up to 10 cells. The cells are cylindrical, isodiametric, sometimes barrel-shaped, and 1.8–6.8 × 2.3–4.3 µm in size. The cells in the trichome are light, blue-green, uniform, and not granular. The heterocysts are terminal, unipolar, spherical or elongated, yellow-green, and 3.0–5.9 × 2.3–3.2 μm in size. The akinetes are mainly cylindrical, oval, large (5.1–7.7 × 2.6–3.4 μm), light-brown, yellow, granular, and adjacent to the heterocysts. Sometimes there are two akinetes at one end of the trichome. Reproduction is by binary cell division, in one plane only. Cultural characteristics: On a solid medium they form slimy colonies, first green, and then brownish in colour. They grow well at a temperature of 25–30 °C on Allen and Bold media in the light at an initial pH of 7. Old cultures (10–15 days of cultivation) contain both heterocysts and akinetes. Culture J-8 exhibited several morpho-characteristics identical to *Cylindrospermum badium* [41], including similar size dimensions of trichomes, vegetative cell, heterocysts, and akinete, and a flattened exospore. Therefore, the isolate J-8, after preliminary morphological identification, was assigned to cyanobacteria, class *Hormogeneae*, order *Nostocales*, genus *Cylindrospermum*, and species *Cylindrospermum badium*.Culture K-31. The cells are cylindrical, barrel-shaped or spherical, pale- or light-blue- or olive-green, with gas bubbles (vacuoles) or without, but sometimes with granular contents. The terminal cells are slightly elongated, not vacuolated. The heterocysts are intercalary, solitary, spaced apart, oval, sometimes spherical, and usually slightly larger than the vegetative cells. The development of heterocysts was observed when the medium was exhausted (on the 8–15th day of cultivation). The akinetes are spherical, solitary, and located in the middle between two heterocysts. This cyanobacterium reproduces by hormogonia. Cultural characteristics: On solid culture medium, they form mucilaginous colonies, on liquid culture medium, they precipitate mucilaginous amorphous tangles, and the walls of the flasks are fouled by the culture. They were isolated on Allen’s culture medium and grow well on Bold’s, Chu-10, and BG -11 media. The development of heterocysts was observed when the medium was emptied at 8–15 days of cultivation. Optimal growth on this medium was observed at a temperature of 25–28 °C [42,43]. Morphological identification of the isolate shows that the isolate belongs to cyanobacteria, class *Hormogeneae*, order *Nostocales*, family *Nostocaceae*, genus *Trichormus*, species *Trichormus variabilis*.Culture J-1. Filamentous cyanobacteria. The cells formed straight, sometimes slightly curved, immobile trichomes up to 10–15 µm in length. At the same time, bushiness (bushy growth) is observed in some places due to the false branching of the trichomes, which often occurs at the sites of heterocyst formation. The cells in the trichomes are often cylindrical, isodiametric, and 1.3–3.2 µm in size. The cells in the trichome are light, green, uniform, and non-granular. The heterocysts are intercalary, spherical or elongated, yellow-green, and 3–3.2 × −5.9 µm in size. The presence of akinetes was not observed. Reproduction is by binary cell division, only in one plane. Cultural characteristics: Yellow-green colonies form on a solid medium. They grow well at a temperature of 25–30 °C on BG-11, Allen, and Bold media in the light, at an initial pH of 7. According to taxonomic affiliation, they are assigned to cyanobacteria, class *Hormogeneae*, order *Nostocales*, family *Scytonemataceae*, genus *Tolypothrix*, and species *Tolypothrix* sp. [44].Culture SH-12. A filamentous cyanobacterium with a mucilaginous sheath. No active movement of the trichomes was observed, but some rocking of the trichome tips was. The width of the cell is 2.6–5 µm, i.e., it is two- to three-fold shorter than the length. They reproduce by binary cell division and in one plane only. Cultural characteristics: On a liquid medium, the suspension is blue-green in mass, slightly mucous-like, and precipitates into an amorphous precipitate that is easily stirred up. The strain develops regardless of the season and remains axenic during storage. This cyanobacterium is isolated on Gromov’s culture medium and grows well on Allen’s and BG-11 media. Blue-green trichome balls form on the surface of Gromov’s agar medium. Optimal cultivation conditions are on Gromov’s medium at an incubation temperature of 25–28 °C, medium pH was 7.5–8. Based on the morphological characteristics, which are congruent with the description reported in previous literature, the isolated strain was identified as *Oscillatoria brevis* [45,46]. According to the systematic position, they belong to the cyanobacteria, class *Hormogeneae*, order *Oscillatoriales*, genus *Oscillatoria*, and species *Oscillatoria*
*brevis*.

The isolated pure cultures are stored in the biotechnology laboratory of the Department of Biotechnology of Al-Farabi Kazakh National University in the Collection of microalgae and cyanobacteria CCMKazNU (Culture Collection of microalgae, Al-Farabi Kazakh National University).

### 3.2. Genetic Identification of Isolated Strains and Phylogenetic Tree Construction

For the molecular identification of the isolated strains, DNA fragments of different lengths were obtained. These DNA fragments were amplified with 16 rRNA universal primers, and similar strands were identified in the NCBI BLAST online database.

Based on the data obtained, eight entries in the chain with high similarity were selected, and several alignments were performed using the MUSCLE program. The phylogenetic tree was constructed using MEGA 6 software based on the evolutionary distance calculated using the neighbour-joining method based on the Kimura-2-Parameter algorithm. The statistical analysis of the tree topology was carried out using a loading analysis of 1000 repeated samples [47].

The phylogenetic tree consists of four main clusters and one group, each represented mainly by different species of *Cylindrospermum*, *Anabaena*, *Tolypothrix,* and *Oscillatoria. Chlorella antarctica* strain CCAP 211/45 (KU291887.1) was used as an outgroup. 

Figure 2 shows the phylogenetic relationships of the neighbouring strains and the topology of the tree, which is the basis for the sequential comparison of the taxa.

The SH-12 strain (MH341423) was 99.76% close to *Oscillatoria tenuis* (JN382222.1) and Uncultured *Oscillatoria sp.* clone GU3-6 (JN382233.1). The phylogram showed that strain J-14 belongs to *Nostoc* genus, similar to *Nostoc muscorum* with accession number HF678508.1 (sc. = 2326; sim. = 99.38%), and *Nostoc sp.* with accession number KM019921.1 (sc. = 2298, sim. = 99.45%). Strain J-1 showed 99.33% affinity to *Tolypothrix tenuis* (KM019944.1) and 99.41% similarity to *Scytonema mirabile* (KM019943.1) showed. 1354 nucleotide sequences were obtained with a combination of two additional primers. The J-8 chains were identified with universal primers 27f and 1492r and showed a close affinity with *Cylindrospermum* sp. BTA1113 (KM652β630.1) (sc. = 785, sim. = 94.36%). The affinity of other neighbouring species was 90.47%, such as *Desmonostoc salinum* (sc. = 702), *Desmonostoc sp.* (90.36%). The results showed that strain K-31 was closely related to *Nostoc* sp. SAG 34.92, *Anabaena variabilis* BTA with 96.62% and *Trichormus variabilis* 94.21% similarity. Despite the fact that the percentage of similarity with *Anabaena variabilis* is greater, according to morphological features (the location of akinetes in relation to heterocysts), strain K-31 is assigned to the genus *Trihormus*. The genus *Trihormus* was isolated from the genus *Anabaena* within the subfamily Nostocoidae. According to the information available in the literature, the genus *Trichormus* includes *A. variabilis*, *A. azollae* Strasb., *A. doliolum* Bharadw, and a number of other species, and thus the names *A. variabilis* and *T. variabilis* (Kütz.) Kom. and Anagn. at one time were considered synonyms [48]. However, despite this, the genus *Trichormus* is characterized by the formation of akinetes from vegetative cells in the middle, between two heterocysts, while in *Anabaena* cells, akinetes are formed nearby with heterocysts, and their growth is directed away from the heterocysts. In the K-31 strain, akinetes were located in the middle between two heterocysts at an equal distance; therefore, this strain was assigned to the genus *Trichomus*. According to the research results, a genetic analysis of five cyanobacterial cultures was performed based on phylogenetic analysis, their close groups were identified, and phylogenetic trees were formed. The resulting nucleotide chains were uploaded to the GenBank database (NCBI) and accession numbers were assigned. Through phylogenetic and morphological analyses, the isolated new cyanobacterial isolates were identified as follows: *Trichormus variabilis* K-31 (MZ079356), *Cylindrospermum badium* J-8 (MZ079357), *Nostoc* sp. J-14 (MZ079360), *Oscillatoria brevis* SH-12 (MZ090011), and *Tolypothrix tenuis* J-1 (MZ079361).

### 3.3. Investigation of the Productivity of Isolated Cyanobacterial Strains on a Nitrogen-Free Medium

The next step was to investigate the ability of the isolated cyanobacteria to grow on a nitrogen-free nutrient medium (Allen_0_) while determining the formation of heterocysts on the filament and the nitrogenase activity of the strains under these conditions. For this purpose, the cultures of four heterocystous cyanobacterial strains were grown on the nutrient medium Allen_0_, and the number of heterocystous (HC) and vegetative cells (VC) within a filament, as well as the growth rate coefficient of the cultured cyanobacteria, were recorded daily. After 14 days, the biomass of the cyanobacteria was separated from the culture fluid and dried and, thereafter, the yield of dry biomass was determined. The results obtained are summarised in Table 1.

In general, it can be stated that all strains examined were able to grow on a nitrogen-free medium. However, compared to the control, a significant prolongation of the lag phase of culture growth on a nitrogen-free medium was observed. In the course of cultivation, the growth rate coefficient of the examined strains reached different values, whereby the most active growth was observed in the representatives of the order *Nostocales*—the strains *Trihormus variabilis* K-31 and *Nostoc* sp. J-14—whose yield of dry biomass at the end of the experiment was 1.06 ± 0.09 and 1.01 ± 0.09 g L^−1^, respectively.

The results obtained indicate a significant difference in the productivity of the representatives of the order *Nostocales* on a nitrogen-free medium: the highest productivity among the strains studied was observed in *Trihormus variabilis* K-31 and *Nostoc* sp. J-14. The ability to grow on a medium without nitrogen in all the cultures studied already indicates the presence of nitrogen-fixing activity in isolated cultures of cyanobacteria. The registration of the number of heterocysts showed that an increase in the number of heterocysts was observed in all cultures on a nitrogen-free medium, despite a slight decrease in the growth rate. After 9 days of culturing the strains on a nitrogen-free medium, it was found that there were no significant differences between the cultures grown on both Allen nutrient medium and Allen_0_ nitrogen-free medium. In the experiment in Allen nutrient medium, three strains were characterised by the active formation of heterocysts. It was found that the cells of *Trihormus variabilis* K-31 formed 5 HC/filament, *Nostoc* sp. J-14, 3 HC/filament, and the strains *Cylindrospermum badium* J-8 and *Tolypothrix tenuis* J-1 released a maximum of 4 HC/filament. These values are about two- to three-fold higher than in the control, where the number of heterocysts correlates positively with cell density or biomass (Figure 3).

Then, five isolated cyanobacterial strains were tested for nitrogenase activity, including *Oscillatoria brevis* SH-12. A cyanobacterial strain that does not form heterocysts (*Synechocystis* sp. PCC 6803) was used as a control. The results of gas chromatography showed that all strains of nitrogen-fixing cyanobacteria exhibited nitrogen activity in the acetylene medium. However, their ability to reduce acetylene to ethylene was found to vary. According to the results, among the species studied, the strain *Oscillatoria brevis* SH-12 showed no nitrogenase activity, as expected, while the strain *Trichormus variabilis* K-31 had the highest ethylene content (3.57 ± 0.26 nmol ethylene mg^−1^ DW h^−1^). In addition, the strains *Nostoc* sp. J-14 (2.82 ± 0.07 nmol ethylene mg^−1^ DW h^−1^) and *Tolypothrix tenuis* J-1 (2.22 ± 0.06 nmol ethylene mg^−1^ DW h^−1^) showed good results. In the strain *Cylindrospermum badium* J-8, this indicator was 1.36 ± 0.05 nmol ethylene mg^−1^ DW h^−1^ (Figure 4).

Thus, based on the experiments, it was found that four out of five isolated cyanobacterial strains have some ability to fix nitrogen. The best results were obtained for representatives of the order *Nostocales*. The strains *Trihormus variabilis* K-31 and *Nostoc* sp. J-14 had the highest nitrogen fixation activity.

### 3.4. Influence of Mo and Fe on Photosynthetic and Nitrogenase Activities of Isolated Cyanobacterial Strains

The effect of the investigated biogenic metals on the photosynthetic activity of the isolated strains was assessed using some fluorescence parameters and the biomass pigment composition. Using an M-PEA-2 fluorimeter, which allows for the simultaneous measurement of induction fluorescence curves and the redox conversions of photosystem components in the microsecond range, the effect of different concentrations of molybdenum (Mo) and iron (Fe) on the maximum quantum yield of PSII of the primary photochemical reactions *Fv*/*Fm* (φ_Po_) was investigated as a first step. This indicator was determined at the time of the peak of nitrogenase activity in a nitrogen-free medium (i.e., according to preliminary data, this is about 24 h).

According to the data obtained, the maximum quantum yield of PSII of the primary photochemical reactions *Fv*/*Fm* correlated positively with an increase in the Fe concentration in the medium in both strains. In *Trihormus variabilis* K-31, a linear increase in the *Fv*/*Fm* parameter was observed with increasing Fe concentration, from values of 0.31 to 0.58 at 1 to 10 µmol Fe concentrations. With a further increase in the metal concentration to 15 µmol, a sharp decrease in the *Fv*/*Fm* value to 0.28 is observed (Figure 5).

The strain *Nostoc* sp. J-14 showed a similar pattern and differed from *Trichomus variabilis* K-31 with lower values. In general, a clear relationship between Fe concentration and PSII photochemical quantum yield was also observed. Thus, *Fv*/*Fm* values for *Nostoc* sp. J-14 were 50% lower under conditions of low metal concentration (1 µmol) and were 0.26 compared to conditions of medium enrichment with 10 µmol Fe (0.50).

It should be noted that in the experiments with Mo, despite its stimulating effect on nitrogenase activity in cyanobacteria, the parameters of the maximum quantum yield of PSII of the primary photochemical reactions *Fv*/*Fm* were significantly lower compared to Fe.

Thus, within the investigated Mo concentrations, the maximum value of the *Fv*/*Fm* parameter was found at a Mo concentration of 1 µmol. For the strain *Trihormus variabilis* K-31, this value was 0.42, for *Nostoc* sp. J-14, 0.32. The ratio of *Fv*/*Fm* values for both strains at different Mo and Fe concentrations in comparison is shown in Figure 6.

To investigate pigment composition, the strains were cultured for 14 days in Allen medium without nitrogen, after which the resulting biomass was analysed for chlorophyll *a*, carotenoid, and phycocyanin content. Metal concentrations were selected based on preliminary studies. Since the yield of dry biomass was different in the different variants of the experiment, the content of pigments was recalculated for 1 mg g^−1^ DW. The results obtained are summarised in Table 2.

The obtained results show that the metals concentrations optimal for nitrogen fixa-tion affect the process of photosynthesis in different ways, especially the pigments content. As expected, the addition of Fe to the medium compared to Mo increases the photosyn-thetic activity of both cyanobacterial strains under nitrogen deficiency. Thus, considering Trichomus variabilis K-31, the content of chlorophyll a in control A (medium without nitro-gen) was 0.29 ± 0.015 mg g^−1^ DW being significantly higher in the presence of 10 μmol Fe (0.46 ± 0.018 mg g^−1^ DW). In control B (Allen medium with standard composition) the content of chlorophyll a was 0.45 ± 0.014 mg g^−1^. The content of carotenoid in the presence of Fe was amounted to 0.23 mg g^−1^, which corresponded to control B—0.22 ± 0.011 mg g^−1^, while this indicator on the medium without nitrogen was 0.14 ± 0.012 mg g^−1^. A slightly different picture was obtained for phycocyanin. The best accumulation of phycocyanin was observed in the presence of Fe (0.58 ± 0.018 mg g^−1^) compared to the controls and the Mo variant.

The influence of the selected Mo and Fe concentrations on the nitrogenase activity of *Trichormus variabilis* K-31 and *Nostoc* sp. J-14 was determined after 1, 2, 4, 8, 24, and 32 h.

The results obtained proved the stimulating effect of the investigated metals on the nitrogenase activity of the cyanobacteria.

Thus, in *Trichomus variabilis* K-31, when exposed to 1 μmol Mo, the maximum accumulation of C_2_H_4_ was observed after 24 h and 5.2 ± 0.6 nmol ethylene mg^−^^1^ DW h^−^^1^, which is 48% higher than in the control (Figure 7). According to the data obtained, nitrogenase activity was high up to 24 h, and the amount of ethylene decreased in the following hours. This, in turn, is closely related to a decrease in glycogen stores. As you know, nitrogen fixation is an energy-rich process in which the nitrogenase releases three identical products simultaneously. Sufficiently good results were obtained with Fe addition, and the maximum oxidation of C_2_H_4_ showed 4.2 ± 0.8 nmol ethylene mg^−^^1^ DW h^−^^1^ at a concentration of 5 µmol after 24 h of the experiment.

It should be noted that nitrogenase activity was significantly lower in the control. The content of ethylene reached 1.6 ± 0.5 nmol ethylene mg^−^^1^ DW h^−^^1^ in the first hours of the experiment, which increased to a maximum point of 3.6 ± 0.5 nmol ethylene mg^−^^1^ DW h^−^^1^ after 24 h of the experiment.

In the experiment with the strain *Nostoc* sp. J-14, the isolate also showed ethylene activity when exposed to the investigated metals; however, the oxidation of C_2_H_4_ was comparatively lower than that of *Trichormus variabilis* K-31. At 1 µmol Mo, 1.5 ± 0.2 nmol ethylene mg^−^^1^ DW h^−^^1^ C_2_H_4_ was measured after 1 h. A high rate of oxidation of C_2_H_2_ to C_2_H_4_ was observed after 24 h (4.4 ± 2.6 nmol ethylene mg^−^^1^ DW h^−^^1^), which is 57% more than in the control. Under the influence of Fe, the maximum oxidation of C_2_H_4_ at a concentration of 10 µmol after 24 h of the experiment was 3.8 ± 0.7 nmol ethylene mg^−^^1^ DW h^−^^1^ (35% more than in the control), in the following hours, a noticeable decrease in this indicator was observed (Figure 8).

According to the results obtained, it was found that the examined concentrations of each metal have a stimulating effect on nitrogenase activity and operation.

## 4. Discussion

Soil erosion and the restoration of arable land after the excessive use of chemical fertilisers can be improved by switching to organic farming, and restoration is generally ensured by the use of biofertilisers as well as by restoring the biological components of the soil. Cyanobacteria, fulfilling many different functions in the soil ecosystem, can be considered to be as promising objects in agrobiotechnology to produce biofertilisers. It is known that the absorption activity of plant root systems leads to an increased outflow of nitrogenous metabolites from the soil. The replenishment of this pool can be maintained by the high activity of the cyanobacterial nitrogenase complex. The study of the phototrophic microorganism biodiversity in the rice fields of Kazakhstan and the isolation of nitrogen-fixing cyanobacteria from water, soil, algal mat, and plant samples was carried out to search for active strains of great practical interest.

The observed rich biodiversity of microalgae and cyanobacteria in the algoflora of the rice fields of the Kyzylorda region, with the characteristic seasonality of the species encountered, is certainly primarily related to the hydrological regime and soil properties, as well as the rice cultivation phase, the use of agrotechnical methods of rice cultivation, and seasonal temperature phenomena. Thus, during the period characterised by increased rice development, increasing shading of the water surface is observed, leading to a noticeable change in the ratio of the green species number, and they are replaced by shade-tolerant species of blue-green algae and diatoms, which form continuous films on the bottom of the rice fields.

The study of the morphological characteristics of the isolated strains J-14, K-31, J-8, J-1, and SH-12 allowed us to determine their affiliation to the orders *Nostocales* and *Oscillatoriales*, and through the analysis of phylogenetically similar species, they were identified as follows: *Trichormus variabilis* K-31 (MZ079356), *Cylindrospermum badium* J-8 (MZ079357), *Nostoc* sp. J-14 (MZ079360), *Oscillatoria brevis* SH-12 (MZ090011), and *Tolypothrix tenuis* J-1 (MZ079361).

An examination of the ability of the isolated heterocystous cyanobacteria to grow on a nitrogen-free nutrient medium, the heterocysts formation on a filament, and the nitrogenase activity of the strains under these conditions showed that all the strains examined could grow on a nitrogen-free medium. The results indicate a difference in the productivity of the representatives of the order *Nostocales* on a nitrogen-free medium. The cells of *Trichormus variabilis* K-31 and *Nostoc* sp. J-14 showed the highest productivity among the strains studied. The ability of all cultures examined to grow on a medium without a nitrogen source already indicates the presence of nitrogen-fixing activity in the isolated cyanobacterial cultures. However, compared to the control, a significant prolongation of the lag phase was observed on a nitrogen-free medium, which is apparently related to the adaptation of the cells to these conditions and the subsequent differentiation of some vegetative cells into heterocysts, with a corresponding shift in all phases of microbial population development. As is known, the formation of heterocysts begins after about 20 h of nitrogen starvation. This is because the genes for nitrogen fixation (*nif*) are expressed about 18–24 h after nitrogen deprivation [49].

Thus, the registration of the number of heterocysts in our studies showed that an increase in the number of heterocysts was observed in all cultures on a nitrogen-free medium, despite a slight decrease in the growth rate. Our results confirm the influence of nitrogen availability for cyanobacterial cells on the frequency of heterocyst formation and the species specificity of this factor. According to data in the literature, in the absence of a nitrogen source in the culture medium, 5–10% of the cells of the filaments of nitrogen-fixing cyanobacteria differentiate into heterocysts within about 24 h [50]. Heterocysts supply the vegetative cells with fixed nitrogen and receive carbohydrates from them in return [51,52]. This mutual exchange of balanced nutrients is regulated at the gene level to ensure their optimal growth and adaptation for long-term survival. At the same time, the genes responsible for heterocyst differentiation control their ratio to the number of vegetative cells on the filaments, ensuring an adequate balance between fixation by vegetative cells and nitrogen fixation by heterocysts. Thus, the NtcA-regulated *pat*D gene is reported to control the ratio of heterocysts relative to vegetative cells on the filaments of *Anabaena* sp. PCC 7120 [53]. It should be noted that according to our results, the frequency of heterocyst formation was even significantly higher under nitrogen deficiency in the medium in some strains. Thus, in the filamentous cyanobacterium *Trichormus variabilis* K-31, 15–20 cells differentiated into heterocysts with an average initial frequency of 8.2 ± 1.8 vegetative cells per heterocyst, indicating the species specificity of this factor.

Subsequently, five isolated cyanobacterial strains were tested for their nitrogenase activity, including *Oscillatoria brevis* SH-12. Although this strain does not form heterocysts, according to recent data, the ability to fix nitrogen is characteristic not only of heterocystous strains, but also of those in which the processes of nitrogen molecule reduction and photosynthesis are separated in time and regulated by a circadian rhythm [54]. Based on the experiments, it was found that four of the five isolated cyanobacterial strains have some ability to fix nitrogen. Consequently, the best results were obtained for representatives of the order *Nostocales*. The strains *Trihormus variabilis* K-31 and *Nostoc* sp. J-14 had the highest nitrogen fixation activity. In contrast to the existing data on the nitrogenase activity of species of the genus *Oscillatoria* [55], we could not detect nitrogenase activity in the cells of *Oscillatoria brevis* SH-12 in our study.

The activation of the process of nitrogen fixation is not only closely linked to the pool of nitrogen in the medium, but also depends on various external factors, including the content of various heavy metals. The process of nitrogen fixation is ensured by the work of a complex enzyme with several subunits and substrates called nitrogenase, which contains a metal in its active centre, and undoubtedly depends on their concentrations in the medium. Depending on the metal contained in the active centre, three types of nitrogenase are distinguished: Mo-nitrogenase (Mo), V-nitrogenase (V), and Fe-nitrogenase (Fe). In this context, it was advisable to investigate the effects of different Mo and Fe concentrations on the nitrogenase activity of the isolated strains and to select the optimal concentrations. For this purpose, two productive strains of the cyanobacteria *Trihormus variabilis* K-31 and *Nostoc* sp. J-14, which were proven to be the most active in previous studies, were selected.

The results obtained evidenced the stimulating effect of certain concentrations of the investigated metals on the cyanobacterial nitrogenase activity. The best results in the study of nitrogenase activity of *Trichormus variabilis* K-31 were obtained when it was exposed to Mo at a concentration of 1 μmol and 10 μmol Fe.

Thus, according to the research results, it was found that a certain amount of each metal studied has a stimulating effect on the operation of nitrogenase. According to the available literature data, nitrogenase responds to the availability of Fe and Mo in the media [56]. In our studies, the best results were obtained for both cyanobacterial cultures in the variants of the experiment with Mo at a concentration of 1 µmol and 10 µmol Fe. The results obtained confirm the presence of both types of nitrogenases in the cyanobacterial isolates studied. It is known that in such cases when the Mo concentration in the medium becomes limited, the work of alternative nitrogenases is expressed [57,58]. Thus, when Mo availability is higher, an increased efficiency of Mo nitrogenase is observed compared to V- and Fe-dependent enzymes [59]. In contrast to another metal studied, Mo is involved in nitrogen metabolism only in cyanobacterial cells and, although the cellular requirement for Mo is very low, this metal plays an essential role in the functioning of nitrogenase, the key element in nitrogen fixation [60]. A deficiency of Mo, which is part of the Fe-Mo cofactor of the enzyme, leads to a decrease in nitrogen fixation despite higher heterocyst and pigment concentrations [61]. When molybdate is excluded from the medium and vanadate is absent, the nitrogenase genes *nif*1 and *vnf* are expressed in heterocysts. However, since they cannot produce nitrogenase, these nitrogen-depleted cells produce a very high frequency of heterocysts and overexpress nitrogenase genes [62,63]. Our observations suggest the stimulatory effect of Mo on the activation of the enzyme. Increasing the Mo concentration to 1 µmol increases the activity of nitrogenase. The results obtained correlate with the data available in the literature. Thus, according to the data presented, the growth rate of *A. variabilis* was almost the same under the influence of V and Mo, but the catalytic efficiency of the alternative nitrogenase was significantly lower than that of the MoFe-nitrogenase [64]. It was also found that the activity of VFe-nitrogenase in *Azotobacter* was about 1.5-fold lower than that of MoFe-nitrogenase under the same cultivation conditions [65].

Iron plays an equally important role in the process of nitrogen fixation. As is known, nitrogen fixation is an Fe-dependent process, since the nitrogenase complex contains 38 Fe atoms per holoenzyme [66]. Among the strains examined, nitrogenase activity was significantly higher after Fe addition in the culture of *Trichormus variabilis* K-31 (4.2 nmol ethylene mg^−^^1^ DW h^−^^1^) compared to *Nostoc* sp. J-14 (3.8 nmol ethylene mg^−^^1^ DW h^−^^1^). At the same time, among the concentrations that were studied, 10 µmol was optimal, a lower concentration (1 µmol) was not effective enough, and a high concentration (15 µmol) had a toxic effect, leading to negative changes in culture growth and low nitrogen fixation. The need for Fe in nitrogen reduction processes is related to the fact that these processes require Fe both in enzymes and for the ferredoxin-mediated provision of the restorative energy of photosynthesis or respiration [67]. Nitrogen fixation metabolism requires the synthesis of Fe and Fe-Mo subunits. In nitrate reduction (either from nitrogen fixation or from an external source), Fe is directly involved in nitrite reductase and, as a ferredoxin cofactor, for nitrate reductase. An iron limitation has been reported to reduce nitrate reductase levels in algae [68]. The entire pathway of nitrogen reduction (from either dinitrogen or nitrate) requires a flux of reducing agents from photosynthesis, resulting in an increased cellular demand for these Fe compounds [69]. It has been reported that nitrogen-fixing cyanobacteria require up to 10-fold more Fe than the same species growing at the same rate on nitrates [70,71].

The chosen concentrations of these metals were optimal for both nitrogen fixation and photosynthesis. At the same time, as expected, the addition of Fe in a nitrogen-free medium compared to Mo increased the photosynthetic activity of both cyanobacterial strains compared to control A (Allen medium without N source). It is known that Fe is necessary as a redox catalyst in photosynthesis and nitrogen assimilation for electron transfer and the formation of NADPH. Iron deficiency can reduce the concentration of chlorophyll and carotenoid in the cells of phototrophic microorganisms [5,72].

## 5. Conclusions

The broad spectrum of activity of cyanobacteria and their importance for the biosphere and the economy of humankind are good reasons to study and understand their diversity, conservation, and use in the interest of our society. Nitrogen-fixing cyanobacteria, for example, which have the simplest nutrient requirements in nature, make an enormous contribution to the nitrogen cycle and, thus, to the greening of agriculture. Maintaining an equilibrium of nitrogen input into the soil as a result of the process of nitrogen fixation is crucial for maintaining soil fertility over long periods. Although a large amount of data has been collected today on the diversity of cyanobacteria in aquatic and soil ecosystems, the search for new strains of cyanobacteria that have great potential for agricultural biotechnology and the replenishment of microbial collections is very important and relevant. This paper presents the results of isolating new nitrogen-fixing cyanobacterial strains and studying their photosynthetic and nitrogenase activities, as well as the prospects for stimulating their nitrogenase activity with two metals important for the process of nitrogen fixation. The optimal concentrations for cyanobacterial growth and stimulation of nitrogenase activity were determined.

## Figures and Tables

**Figure 1 cells-11-00904-f001:**
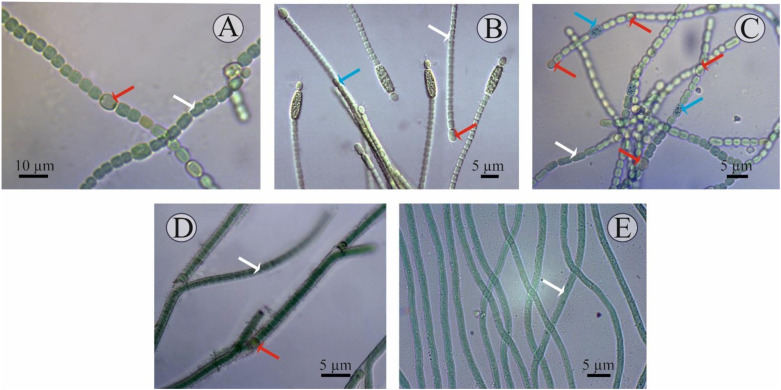
Microscopic images of the cyanobacterial isolates: (**A**) *Nostoc* J-14, (**B**) *Cylindrospermum* J-8, (**C**) *Trichormus* K-31, (**D**) *Tolypothrix* J-1, and (**E**) *Oscillatoria* SH-12. Formation of the different cell types are marked in arrow colours: white—vegetative cells; red—heterocysts; blue—akinetes.

**Figure 2 cells-11-00904-f002:**
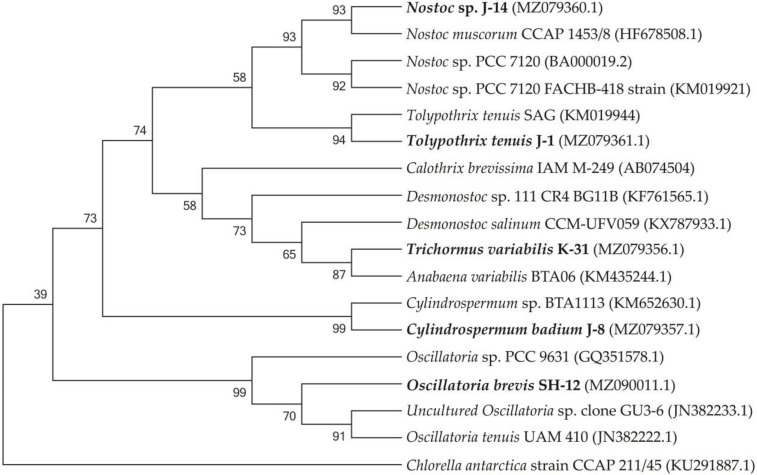
Isolated cyanobacterial strains and phylogenetic tree of neighbouring species.

**Figure 3 cells-11-00904-f003:**
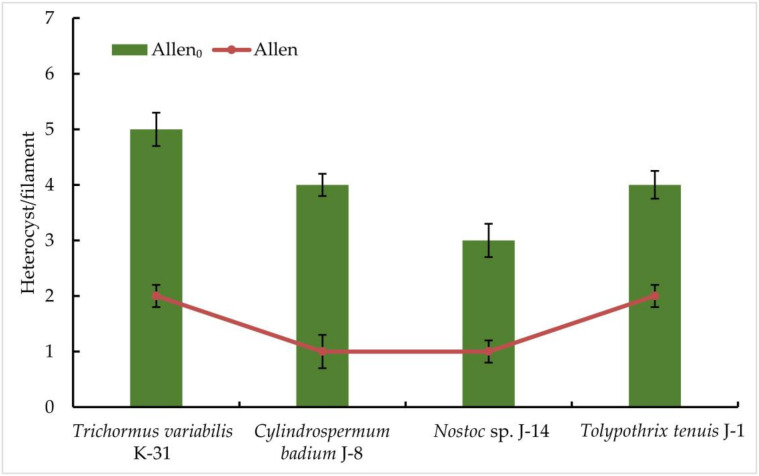
The frequency of heterocyst formation during nitrogen starvation in isolated cyanobacterial strains. The species were grown in the Allen medium for 14 days. The biomass was collected and transferred to the Allen_0_ medium. Then, heterocyst formation was studied under a light microscope. Different letters on the bar and line chart indicate a significant difference at *p* < 0.05.

**Figure 4 cells-11-00904-f004:**
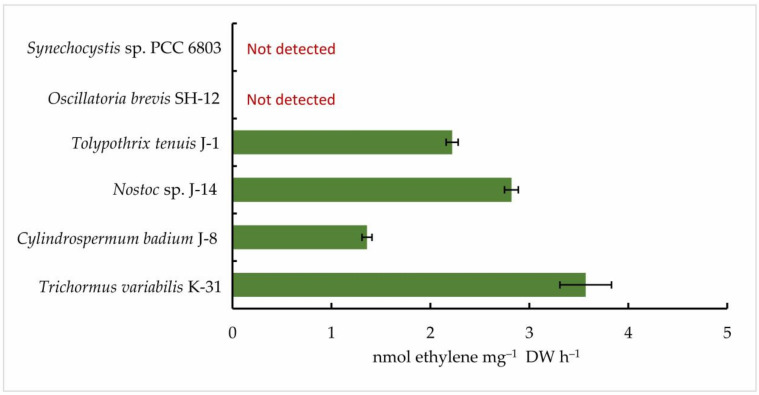
Nitrogenase activity of isolated cyanobacterial strains. The cells were cultured until the beginning of the stationary phase of the cells. Next, collected biomass was transferred to the Allen_0_ nitrogen-free medium. After 24 h, the concentration of ethylene was measured. The data are presented as mean +/− SE (*n* = 3). Different letters on the bar indicate a significant difference at *p* < 0.05.

**Figure 5 cells-11-00904-f005:**
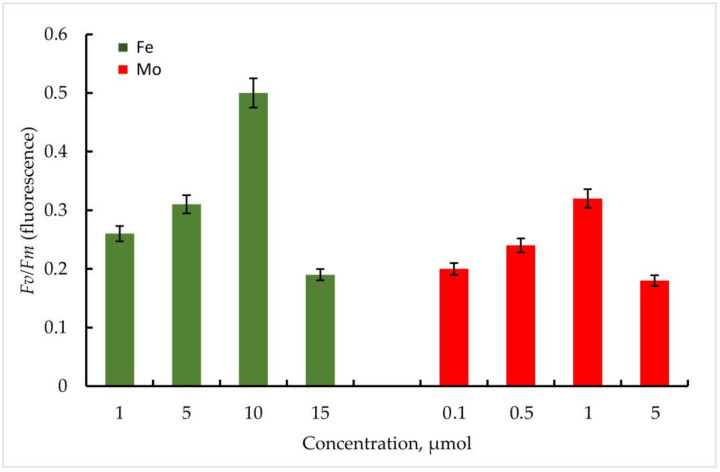
Maximum quantum yield of PSII (*Fv*/*Fm*) of primary photochemical reactions at different Mo and Fe concentrations for *Trihormus variabilis* K-31. The culture was grown on Allen medium for 7 days. The obtained biomass was transferred to the medium containing different concentrations of Fe (1, 5, 10, and 15 µmol) and Mo (0.1, 0.5, 1, and 5 µmol) metals. Then, the ratio of *Fv*/*Fm* values was measured. The data are presented as mean +/− SE (*n* = 4). Different letters on the bar indicate a significant difference at *p* < 0.05.

**Figure 6 cells-11-00904-f006:**
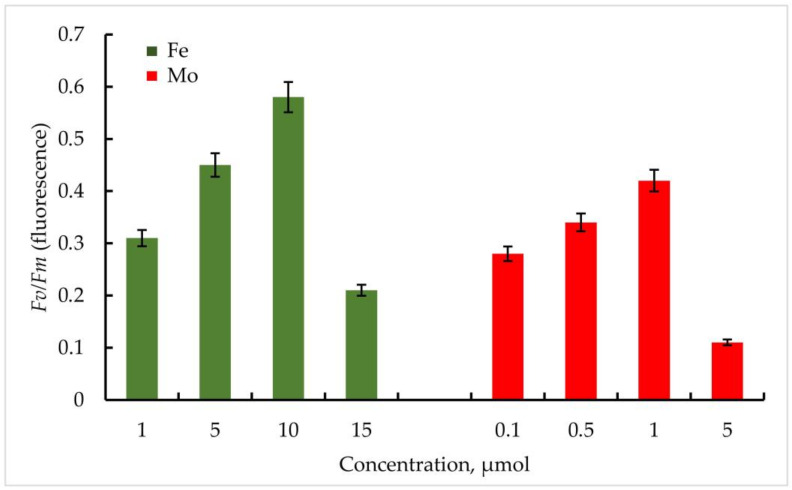
PSII maximum quantum yield *Fv*/*Fm* (φ_Po_) of primary photochemical reactions at different Mo and Fe concentrations for *Nostoc* sp. J-14. The culture was grown on Allen medium for 7 days. The obtained biomass was transferred to the medium containing different concentrations of Fe (1, 5, 10, and 15 µmol) and Mo (0.1, 0.5, 1, and 5 µmol) metals. Then, the ratio of *Fv*/*Fm* values was measured. The data are presented as mean +/− SE (*n* = 4). Different letters on the bar indicate a significant difference at *p* < 0.05.

**Figure 7 cells-11-00904-f007:**
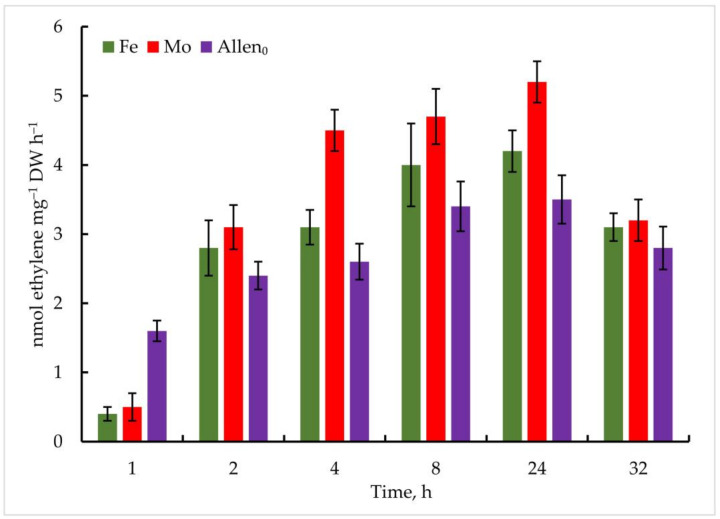
Metal-induced increase in nitrogenase activity in *Trichormus variabilis* K-31 cells. The culture was grown on Allen medium for 7 days. Then, the cells were transferred to media containing Mo (1 µmol) and Fe (10 µmol) at time 0. Acetylene yield was measured at 1, 2, 4, 8, 24, and 32 h after the addition of the Mo and Fe containing salts. The data are presented as mean +/− SE (*n* = 5). Different letters on the bar indicate a significant difference at *p* < 0.05.

**Figure 8 cells-11-00904-f008:**
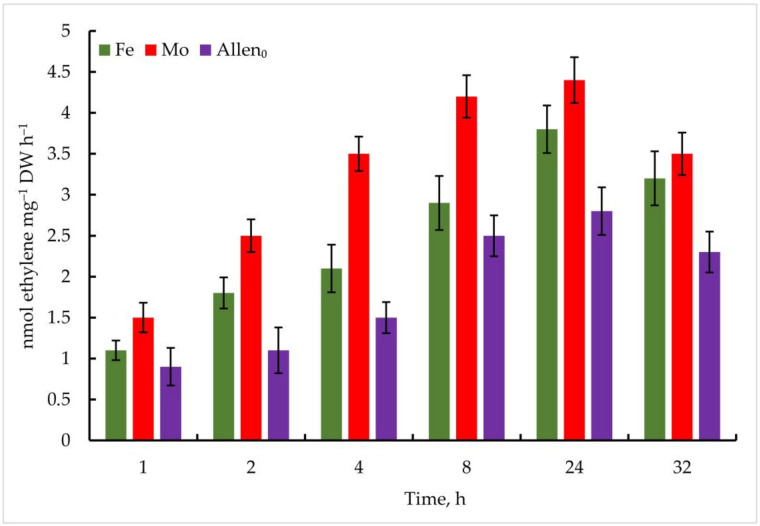
Metal-induced increase in nitrogenase activity in *Nostoc* sp. J-14 cells. The culture was grown on Allen medium for 7 days. Then, the cells were transferred to media containing Mo (1 µmol) and Fe (10 µmol) at time 0. Acetylene yield was measured at 1, 2, 4, 8, 24, and 32 h after the addition of the Mo and Fe containing salts. The data are presented as mean +/− SE (*n* = 5). Different letters on the bar indicate a significant difference at *p* < 0.05.

**Table 1 cells-11-00904-t001:** The productivity results of isolated microalgal strains on a nitrogen-free medium (Allen_0_). The data are presented as mean +/− SE (*n* = 3), *p* < 0.05.

Strain	Growth Coefficient, µ (per Day)	DW, g L^−1^
*Trichormus variabilis* K-31	0.28 ± 0.02	1.06 ± 0.09
*Nostoc* sp. J-14	0.27 ± 0.02	1.01 ± 0.09
*Cylindrospermum badium* J-8	0.19 ± 0.02	0.87 ± 0.08
*Tolypothrix tenuis* J-1	0.24 ± 0.02	0.91 ± 0.07

**Table 2 cells-11-00904-t002:** Pigment composition of cyanobacterial isolates under the influence of the metal. Different letters within one pigment indicate a significant difference at *p* < 0.05.

Strain	Metal	Concentration, µmol	Pigment Content, mg g^−1^ DW		
			Chlorophyll *a*	Carotenoid	Phycocyanin
*Trichormus variabilis* K-31	Mo	1	0.38 ± 0.024	0.18 ± 0.016	0.19 ± 0.011
	Fe	10	0.46 ± 0.018	0.23 ± 0.014	0.58 ± 0.018
	Control A ^1^	NA ^3^	0.29 ± 0.015	0.14 ± 0.012	0.15 ± 0.011
	Control B ^2^	NA	0.45 ± 0.014	0.22 ± 0.011	0.18 ± 0.012
*Nostoc* sp. J-14	Mo	1	0.31 ± 0.021	0.23 ± 0.023	0.13 ± 0.013
	Fe	10	0.37 ± 0.019	0.25 ± 0.016	0.47 ± 0.015
	Control A ^1^	NA	0.31 ± 0.011	0.22 ± 0.011	0.11 ± 0.012
	Control B ^2^	NA	0.35 ± 0.011	0.24 ± 0.019	0.21 ± 0.015

^1^ Allen_0_ medium. ^2^ Allen medium. ^3^ NA— not added.

## Data Availability

The data are available from the corresponding authors upon reasonable request.

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
