# Peer review of "Influence of Mo and Fe on Photosynthetic and Nitrogenase Activities of Nitrogen-Fixing Cyanobacteria under Nitrogen Starvation"

_cells, 2022, doi:10.3390/cells11050904_

Round 1

Reviewer 1 Report

The MS looks good and it is devoted to an important ecological and biotechnological topic associated with cyanobacteria, which constitute 90% of earth biomass. A Lot of data are presented in the MS and much illustrative material. Methods are adequate. English is OK. However, I found some inaccuracies.

Main comments

  1. It is written that Statistical analysis ANOVA was used but we do not see the use of ANOVA in Figs and Table.
  2. Please, explain, why Mo and Fe metals were chosen for research in the MS.
  3. Table 1. The errors are absent in values of DW.
  4. Table 2. Please point out of full names of pigments in the table legend. Besides, it would be good to show reliable differences between pigment contents (Chl a, β Car, and Xc).

Minor comments

  1. I see many Refs to the method of determination of nitrogenase activity (26-29). One can give one-two Refs but describe the method in more detail.
  2. Please, write μmol (photons) m-2 s–1 instead of μmol photon m2 sec–1
  3. One should write everywhere PSII maximal quantum yield instead of maximal quantum yield.
  4. The Figure legends should be elaborated. 
  5. You can cite these articles which is on Fe deficiency from Chlamydomonas: 1) PLoS One. 7(4): e35084. ISSN · 1932-6203;  2) Biochim. Biophys. Acta (bioenergetics) 1862, Issue 1, 148331 ISSN 0005-2728; 3) Photosynth. Res 139(1-3):253-266 doi: 10.1007/s11120-018-0580-2

Author Response

The authors of the manuscript would like to thank you and the reviewers for such a detailed review of our work and the opportunity to revise the manuscript. According to your comments, we did all the necessary corrections and fixed everything. Specific responses to each comment are provided below.

Reviewer # 1

The MS looks good and it is devoted to an important ecological and biotechnological topic associated with cyanobacteria, which constitute 90% of earth biomass. A lot of data are presented in the MS and much illustrative material. Methods are adequate. English is OK. However, I found some inaccuracies.

Answer: Dear Reviewer, thank you for your positive feedback on our manuscript. We are grateful for your help to refine our version. We have corrected this version, taking into account all your pertinent comments below.

Main comments

  1. It is written that Statistical analysis ANOVA was used but we do not see the use of ANOVA in Figs and Table.

Answer: Thanks for the comment. According to your remark, the changes have been made in the figures and tables. Please, kindly find modified tables and figures in the text. Different letters on the bar and line chart indicate a significant difference. Please, kindly see “2.10. Statistical analysis”.

  1. Please, explain why Mo and Fe metals were chosen for research in the MS.

Answer: Thanks for the comment. The activation of the process of nitrogen fixation is closely related not only to the nitrogen content in the environment, but also depends on various external factors, including the content of various heavy metals. Mo and Fe are metals found in the active site of the nitrogenase enzyme. Depending on the metal contained in the active center, three types of multisubunit, multisubstrate, complex nitrogenase enzyme are differentiated:

- Mo-nitrogenase (Mo); - V-nitrogenase (V); - Fe-nitrogenase (Fe).

There is information in the literature about the stimulating effect of these metals on nitrogenase activity. The study of the influence of various concentrations of Mo and Fe was carried out in order to determine their optimal concentrations, which have a positive stimulating effect on the nitrogen-fixing ability of isolated new isolates of cyanobacteria. Please kindly see page 18.

  1. Table 1. The errors are absent in values of DW.

Answer: Thanks for the comment. An error has been added in the dry weight values.

  1. Table 2. Please point out of full names of pigments in the table legend. Besides, it would be good to show reliable differences between pigment contents (Chl a, β Car, and Xc).

Answer: Thanks for the comment. In Table 2, the names of the pigments have been corrected for the pigment composition and possible differences between the contents of the three pigments are shown.

Minor comments:

  1. I see many Refs to the method of determination of nitrogenase activity (26-29). One can give one-two Refs but describe the method in more detail.

Answer: Thanks for the comment. According to your comment, we have reduced the number of references, and tried to describe in more detail the method for determining nitrogenase activity in the research methods section. Please see “2.7. Determination of nitrogenase activity by the acetylene method”.

  1. Please, write μmol (photons) m-2s–1 instead of μmol photon m2 sec–1

Answer: Thanks for the comment. This has been fixed. Please see the black file of the text to figure out changed places.

  1. One should write everywhere PSII maximal quantum yield instead of maximal quantum yield.

Answer: Thanks for the comment. This has been fixed.

  1. The Figure legends should be elaborated. 

Answer: Thanks for the comment. This has been fixed.

  1. You can cite these articles which is on Fe deficiency from Chlamydomonas: 1) PLoS One7(4): e35084. ISSN 1932-6203;  2) Biochim. Biophys. Acta (bioenergetics)1862, Issue 1, 148331 ISSN 0005-2728; 3) Photosynth. Res 139(1-3):253-266 doi: 10.1007/s11120-018-0580-2

Answer: Thanks for the information provided. We reviewed these sources and included them in the discussion of the results.

We cited these articles:

Yadavalli, V.; Jolley, C.C.; Malleda, C.; Thangaraj, B.; Fromme, P.; Subramanyam, R. Alteration of Proteins and Pigments Influence the Function of Photosystem I under Iron Deficiency from Chlamydomonas Reinhardtii. PLOS ONE 2012, 7, e35084, doi:10.1371/journal.pone.0035084.

Devadasu, E.; Pandey, J.; Dhokne, K.; Subramanyam, R. Restoration of Photosynthetic Activity and Supercomplexes from Severe Iron Starvation in Chlamydomonas Reinhardtii. Biochimica et Biophysica Acta (BBA) - Bioenergetics 2021, 1862, 148331, doi:10.1016/j.bbabio.2020.148331.

Devadasu, E.; Chinthapalli, D.K.; Chouhan, N.; Madireddi, S.K.; Rasineni, G.K.; Sripadi, P.; Subramanyam, R. Changes in the Photosynthetic Apparatus and Lipid Droplet Formation in Chlamydomonas Reinhardtii under Iron Deficiency. Photosynth Res 2019, 139, 253–266, doi:10.1007/s11120-018-0580-2.

Reviewer 2 Report

The authors isolated four strains of nitrogen-fixing cyanobacteria from rice fields, and investigated effects of Mo and Fe on the photosynthetic and nitorogenase activities. This kind of study is very important in terms of the search for new strains of cyanobacteria that have great potential for agricultural biotechnology. However, I think that clear description of the results and careful correction of the manuscript, as well as improvement of English expression, are required for publication.

Specific points

  1. Figure 1: The microscopic images should be improved, and the heterocysts and akinetes should be clearly indicated by arrow heads or something, since the identification of the isolated strains is one of main parts of this study.
  2. Figure 2 and genetic identification of isolated strains: The K-31 strain was identified as Trichormus variabilis K-31. In lines 407-409, however, you describe that "strain K-31 was closely related to Nostoc ... and Anabaena .... And according to the morphological characteristics ..., it belongs to the species Anabaena." Also, Figure 2 cannot suggest that K-31 belongs to Trichormus. If you indicate that the strain belongs to Trichormus, you should show the strain is included in the clade of Trichormus. To do this, you should analyze more sequences including several Trichormus Similarly, J-8 was identified as Cylindrospermum badium. However, you just mention that "The J-8 chains ... showed a very close affinity with Cylindrospermum sp." Please explaine why the strain is Cylindrospermum badium by revising the figure and the manuscript.
  3. Table 1: Please add a unit for growth coefficient in the table
  4. Table 2: Xanthophyll content is shown in the table. However, in the method section 9., you describe the measurement of fucoxanthin, which is not included in cyanobacteria. Also, I don't think that the equation for fucoxanthin is correct. I think that the equation may be for phycocyanin and Table 1 may show the phycocyanin content but not the xanthophyll content. Please correct the method or the table, or both of them.

Author Response

The authors of the manuscript would like to thank you and the reviewers for such a detailed review of our work and the opportunity to revise the manuscript. According to your comments, we did all the necessary corrections and fixed everything. Specific responses to each comment are provided below.

The authors isolated four strains of nitrogen-fixing cyanobacteria from rice fields, and investigated effects of Mo and Fe on the photosynthetic and nitorogenase activities. This kind of study is very important in terms of the search for new strains of cyanobacteria that have great potential for agricultural biotechnology. However, I think that clear description of the results and careful correction of the manuscript, as well as improvement of English expression, are required for publication.

Answer: Thank you for your comment. We worked all the text and did correction the introduction and other sections.

Specific points

  1. Figure 1: The microscopic images should be improved, and the heterocysts and akinetes should be clearly indicated by arrow heads or something, since the identification of the isolated strains is one of main parts of this study.

Answer: Thanks for the comment. According to your remark, the work presents clearer photographs and heterocysts and akinetes are indicated by arrows of different colors for better understanding.

  1. Figure 2 and genetic identification of isolated strains: The K-31 strain was identified as Trichormus variabilisK-31. In lines 407-409, however, you describe that "strain K-31 was closely related to Nostoc ... and Anabaena .... And according to the morphological characteristics ..., it belongs to the species Anabaena." Also, Figure 2 cannot suggest that K-31 belongs to Trichormus. If you indicate that the strain belongs to Trichormus, you should show the strain is included in the clade of Trichormus. To do this, you should analyze more sequences including several Trichormus Similarly, J-8 was identified as Cylindrospermum badium. However, you just mention that "The J-8 chains ... showed a very close affinity with Cylindrospermum"Please explaine why the strain is Cylindrospermum badium by revising the figure and the manuscript.

Answer: Among the cyanobacterial genera, Cylindrospermum is unique in that morphological variety appears to be more prominent than sequence divergence [2]. Taxa are significantly more distinguishable by morphology, as the ribosomal sequences shared a high degree of commonality. The phylogenetic tree verified the placement of J-8 strain in Cylindrospermum genus. Based on the morphological similarity to Cylindrospermum badium described in previous literature [3], J-8 was identified as Cylindrospermum badium. In this regard, few lines have been added in section - results.

“strain J-8 exhibited several morphocharacters identical to Cylindrospermum badium (Johansen et al., 2014), including similar size dimensions of trichomes, vegetative cell, heterocysts, akinete and flattened exospore, therefore J-8 is tentatively designated as Cylindrospermum badium .”

  • Komarek, J. & Anagnostidis, K. Modern approach to the classification system of Cyanophytes 4-Nostocales. Arch. Hydrobiol. Suppl. 82(3), AlgologicalStudies/Archiv für Hydrobiologie, Supplement Volumes 56, 247–345 (1989).
  • Johansen, J. R., Bohunická, M., Lukešová, A., Hrčková, K., Vaccarino, M. A., & Chesarino, N. M. (2014). Morphological and molecular characterization within 26 strains of the genus C ylindrospermum (N ostocaceae, C yanobacteria), with descriptions of three new species. Journal of Phycology, 50(1), 187-202.
  • Johansen, J.R., Bohunická, M., Lukešová, A., Hrčková, K., Vaccarino, M.A. and Chesarino, N.M., 2014. Morphological and molecular characterization within 26 strains of the genus C ylindrospermum (N ostocaceae, C yanobacteria), with descriptions of three new species. Journal of Phycology, 50(1), pp.187-202.

Table 1: Please add a unit for growth coefficient in the table

Answer:  Thanks for the comment. Changes have been made. Please see page 7 and 8.

Table 2: Xanthophyll content is shown in the table. However, in the method section 9., you describe the measurement of fucoxanthin, which is not included in cyanobacteria. Also, I don't think that the equation for fucoxanthin is correct. I think that the equation may be for phycocyanin and Table 1 may show the phycocyanin content but not the xanthophyll content. Please correct the method or the table, or both of them.

Answer: Thanks for the comment. It was a mistake in translation of the manuscript. Yes, we have studied the content of phycocyanin. According to your comment, we have made corrections to the research methods and results sections. Please kindly see relevant places.

Reviewer 3 Report

It is better to include to the main results and key conclusions in the abstract instead of the details of the equipment used for the measurements.

The English suffers from numerous stylistic flaws (e.g. L56: …nitrogen molecule restoration---??; L99: algological pure cultures, ---there are many more).

Section 2.1: please specify the geocoordinates of the sampling sites used for the strain isolation.

LL94, 164, 208: please give references for the media recipes.

L142: please give references for the primers and/or their sequences.

Section 2.3.3: please give references for the software used.

Section 2.6: there is no point in presenting the detailed description of the characteristic points of OJIP curve since only Fo and Fm are used thereafter.

L159: …productivity of…biomass---?? (should be: …biomass productivity…?)

L213: why environment is mentioned here: laboratory cultures were studied?

LL216, 217: what was the rationale for selecting these particular metal concentrations and ionic forms?

L238: what was the reason for assaying fucoxanthin which is absent in cyanobacteria?

Section 2.10: the description of the statistical treatment is totally incomprehensible.

Section 3.1: how the eukaryotic biodiversity was estimated? Only the procedures for prokaryotic biodiversity analysis are described in the methods. By the way, the description of the taxonomic analysis is lacking. Although some hints on this are shown in Fig. 2, all this should be explicitly stated.

Fig. 2: the number of sequences of the organisms with known taxonomic identity from open database shown in this phylogenetic tree is clearly insufficient for reliable clustering. The scale bare should be given.

Figs. 5 and 6: to interpret with confidence the observed effect of the metal addition, one should include the control comprised by the cultures in N-repleted medium with the same metal concentrations added.

Table 2: what are the pigment content units — … /mg DW or mg / … DW?

Author Response

The authors of the manuscript would like to thank you and the reviewers for such a detailed review of our work and the opportunity to revise the manuscript. According to your comments, we did all the necessary corrections and fixed everything. Specific responses to each comment are provided below.

It is better to include to the main results and key conclusions in the abstract instead of the details of the equipment used for the measurements.

Answer: Thanks! According to your remark, the abstract of the manuscript has been amended.

The English suffers from numerous stylistic flaws (e.g. L56: …nitrogen molecule restoration---??; L99: algological pure cultures, ---there are many more).

Answer: Thanks for the comment. Stylistic and grammatical errors have been corrected throughout the text .It is done by proofreader.  

Section 2.1: please specify the geocoordinates of the sampling sites used for the strain isolation.

Answer: Thanks for the comment. The coordinates of the sampling sites have been added to the manuscript. Please see page 8.

LL94, 164, 208: please give references for the media recipes.

Answer: Thanks. References to media are included in the research methods section. Please see “2.2. Isolation and cultivation of cyanobacterial strains”.

L142: please give references for the primers and/or their sequences.

Answer: Thanks. References to primers and their sequences are included in the research methods section. You can find the citation on the page 3.

Section 2.3.3: please give references for the software used.

Answer: Thanks for the comment. Links to the software used are provided in the Research Methods section.

Section 2.6: there is no point in presenting the detailed description of the characteristic points of OJIP curve since only Fo and Fm are used thereafter.

Answer: Thanks. According to your comment, corrections have been made in the research methods section.

L159: …productivity of…biomass---?? (should be: …biomass productivity…?)

Answer: Thanks. Grammar errors corrected.

L213: why environment is mentioned here: laboratory cultures were studied?

Answer: Thank you for your comment. This is a translation error. In this case, the nutrient medium was meant. The bug has been fixed.

LL216, 217: what was the rationale for selecting these particular metal concentrations and ionic forms?

Answer: Thanks for the comment. The study of the influence of various concentrations of Mo and Fe was carried out in order to determine their optimal concentrations, which have a positive stimulating effect on the nitrogen-fixing ability and photosynthetic activity of the isolated new isolates of cyanobacteria. These metals are contained in the active site of the nitrogenase enzyme. But at the same time, it must be remembered that these are still heavy metals. Therefore, when choosing the studied minimum and maximum concentrations of metals, we were guided by the data available in the literature [1-7]. At the same time, the study of the effect of significantly higher concentrations of Fe compared to Mo was certainly related to the needs of phototrophic crops. So, in relation to Fe, this need is much higher, since they need it primarily for the processes of photosynthesis and respiration, and then for the process of nitrogen fixation. In addition, it should be noted that the Maximum Allowable Concentration (MAC) for Fe is also higher compared to Mo. As for the selected salts of these metals, these salts were chosen by us because they are part of the standard nutrient medium for the studied cultures of cyanobacteria and in these salts these metals have high bioavailability for cultures. Please see page 17 and 18.

  • Marino, R.; Howarth, R.W.; Chan, F.; Cole, J.J.; Likens, G.E. Sulfate Inhibition of Molybdenum-Dependent Nitrogen Fixation by Planktonic Cyanobacteria under Sea Water Conditions: A Non-Reversible Effect. In Aquatic Biodiversity: A Celebratory Volume in Honour of Henri J. Dumont; Martens, K., Ed.; Developments in Hydrobiology; Springer Netherlands: Dordrecht, 2003; pp. 277–293 ISBN 978-94-007-1084-9.
  • Fernández-Juárez, V.; Bennasar-Figueras, A.; Sureda-Gomila, A.; Ramis-Munar, G.; Agawin, N.S.R. Differential Effects of Varying Concentrations of Phosphorus, Iron, and Nitrogen in N2-Fixing Cyanobacteria. Frontiers in Microbiology 2020, 11.
  • Berman-Frank, I.; Cullen, J.T.; Shaked, Y.; Sherrell, R.M.; Falkowski, P.G. Iron Availability, Cellular Iron Quotas, and Nitrogen Fixation in Trichodesmium. Limnology and Oceanography 2001, 46, 1249–1260, doi:4319/lo.2001.46.6.1249.
  • Kustka, A.; Sañudo-Wilhelmy, S.; Carpenter, E.J.; Capone, D.G.; Raven, J.A. A Revised Estimate of the Iron Use Efficiency of Nitrogen Fixation, with Special Reference to the Marine Cyanobacterium Trichodesmium Spp. (Cyanophyta)1. Journal of Phycology 2003, 39, 12–25, doi:1046/j.1529-8817.2003.01156.x.
  • Tuit, C.; Waterbury, J.; Ravizza, G. Diel Variation of Molybdenum and Iron in Marine Diazotrophic Cyanobacteria. Limnology and Oceanography - LIMNOL OCEANOGR 2004, 49, 978–990, doi:4319/lo.2004.49.4.0978.
  • Glass, J.B.; Axler, R.P.; Chandra, S.; Goldman, C.R. Molybdenum Limitation of Microbial Nitrogen Assimilation in Aquatic Ecosystems and Pure Cultures. Front Microbiol 2012, 3, 331, doi:3389/fmicb.2012.00331.
  • Xu, Y. Molybdenum and Iron Interactions as Micronutrients for Growth of a Freshwater Cyanobacterium, Microcystis Aeruginosa. Electronic Thesis and Dissertation Repository 2015.

L238: what was the reason for assaying fucoxanthin which is absent in cyanobacteria?

Answer: Thanks for the comment. It was a mistake in translation. We have studied the content of phycocyanin. According to your comment, we have made corrections to the research methods and results sections.

Section 2.10: the description of the statistical treatment is totally incomprehensible.

Answer: Thank you for your comment. We have tried to clarify the description of the statistical processing of the results.

Section 3.1: how the eukaryotic biodiversity was estimated? Only the procedures for prokaryotic biodiversity analysis are described in the methods. By the way, the description of the taxonomic analysis is lacking. Although some hints on this are shown in Fig. 2, all this should be explicitly stated.

Answer: Thanks for the comment. The taxonomic analysis of phototrophic eukaryotes presented in the first section of the "Results" chapter is not given in detail, since the main purpose of this study was to isolate and study nitrogen-fixing strains of cyanobacteria. Therefore, a taxonomic study and genetic identification of the isolated isolates of cyanobacteria was carried out, the results of which are presented in the work. The species diversity of phototrophic eukaryotes, determined only by the morphological features of the species encountered in the studied samples, was included in the manuscript in brief for a general presentation of the composition and dominant species of the algoflora of the studied rice fields. According to your comment, in the research methods section, we have included a list of some of the determinants of phototrophic eukaryotes used in the work. Determination of the species composition of phototrophic eukaryotes was carried out according to morphological and cultural properties using various determinants [1, 2].

1 Huynh, M.; Serediak N. Algae Identification Field Guide. Agriculture and Agri-Food Canada. 2006; pp. 40 ISBN 978-1-100-18308-4.

2 Bellinger, E.G.; Sigee, D.C. Freshwater Algae: Identification and Use as Bioindicators; John Wiley & Sons, 2011; pp. 284 ISBN 978-1-119-96432-2.

Fig. 2: the number of sequences of the organisms with known taxonomic identity from open database shown in this phylogenetic tree is clearly insufficient for reliable clustering. The scale bare should be given.

Answer: Thanks for the comment. The 16S rRNA sequences of organisms with known taxonomic identity from open databases only with more than 99% sequence similarity (2 to 3 organisms) were used in combination with compatibility with morphological observations of studied strains in order to avoid identification and clustering errors. If a matching sequence belong to distant variable strain, we have avoided its use in phylogenetic analysis. The revised phylogenetic diagram has been included in the revised manuscript with scale bar.

Figs. 5 and 6: to interpret with confidence the observed effect of the metal addition, one should include the control comprised by the cultures in N-deprived medium with the same metal concentrations added.

Answer: Thanks for the comment. In this case, we set the task of studying the influence of the studied metals on nitrogenase activity in order to determine certain concentrations for its stimulation. As is known, the presence of sources of bound nitrogen in the medium represses synthesis and inhibits the activity of the nitrogenase enzyme; therefore, the medium without a nitrogen source was taken as a control.

Table 2: what are the pigment content units — … /mg DW or mg / … DW?

Answer: Thanks for the comment. Changes have been made.

Round 2

Reviewer 2 Report

The manuscript has been improved, but still the unit of the growth coefficient in Table 1 is hard to understand. The growth coefficient is not "µ (per hour or per day)"?

Reviewer 3 Report

The authors have dealt with the most important shortcomings of their mansucript, the remaining minor issues (just the spell & grammar issue, not the science) can be removed during copyediting.